# Bestiality Law in the United States: Evolving Legislation with Scientific Limitations

**DOI:** 10.3390/ani12121525

**Published:** 2022-06-12

**Authors:** Brian James Holoyda

**Affiliations:** Martinez Detention Facility, Martinez, CA 94553, USA; holoyda@gmail.com

**Keywords:** bestiality, zoophilia, zoophilic disorder, paraphilia, paraphilic disorder, animal sexual abuse, animal cruelty, the Link

## Abstract

**Simple Summary:**

Laws punishing individuals who have sex with nonhuman animals have existed since the earliest written legal codes. In the United States, bestiality has long been prohibited. The rationale for criminalizing sex acts with animals has changed over time and has included moral condemnation, considerations of animal rights and animal welfare, and most recently, a concern about the relationship between animal cruelty and interpersonal violence, colloquially known as the Link. This article reviews the history and current state of bestiality law in the United States. It notes important differences in language, specificity, and potential punishments for offenders depending on the jurisdiction. It also reviews the research basis of the Link between bestiality and interpersonal violence and some risks associated with a reliance on the Link to promote legislative reform.

**Abstract:**

Societies have proscribed bestiality, or sex between humans and nonhuman animals, since the earliest recorded legal codes. In the early American colonies, religious prohibitions against bestiality provided the grounds for punishing those who engaged in such acts. In the 1800′s, Henry Bergh imported the animal welfare approach to the United States, which modernized the legislative treatment of animals in the country. Until recently, however, many laws in the U.S. have been outdated and vague and have utilized moralistic terminology. Since the 1960′s, a growing body of literature has developed suggesting that individuals who harm animals may also interpersonally offend. This concept, known as the Link, has served as a major motivation for advocates to promote new legislation criminalizing bestiality, to modernize old state statutes, and to expand penalties for individuals convicted of having sex with animals. Unfortunately, data supporting the Link between bestiality and interpersonal violence are limited and of questionable generalizability to the broad public. The Link’s weaknesses can assist in guiding further research. This article summarizes the history of bestiality law, the current state of bestiality legislation in the United States, the body of Link-related literature on bestiality and interpersonal violence and other problematic sexual behaviors, and the empirical weaknesses and needs revealed by this legislation.

## 1. Introduction

Human societies have established laws to regulate sexual acts between humans and nonhuman animals, or bestiality, beginning with the earliest recorded legal codes. Though the legal response to bestiality has largely been punitive, the motivations for such regulation have changed over time. Scholars suggest that the earliest laws were established to regulate cultural concerns about community cleanliness [1]. Since then, motivations for law regulating human–animal interactions, including bestiality, have included religious or moral convictions, animal rights concerns, and most recently, the Link, or the hypothesized relationship between animal cruelty and various forms of interpersonal violence [2]. Today, most jurisdictions in the United States proscribe bestiality, but laws vary widely between states [3]. 

Despite the long-standing recognition of human–animal sexual activity and the evolution of legal mechanisms to regulate the behavior, research on bestiality is limited. Research supporting the Link between bestiality and interpersonal violence is sparse, though a growing body of evidence suggests that individuals with a history of bestiality may be at risk of engaging in various other problematic sexual behaviors. Despite the limitations of this data, some animal advocates have promoted the Link as a means of establishing new animal protection laws or expanding current laws [4]. This article reviews the history of the legal prohibition against bestiality, summarizes the status of bestiality law in the United States, describes the research supportive of the Link between bestiality and interpersonal violence, and identifies research gaps and their potential effect on legislation.

## 2. History of Bestiality Law

### 2.1. Early Societies’ Responses

Prior to 12,000 years ago, human beings relied on animals primarily as a source of food and raw materials in hunter-gatherer societies. It was not until the end of the last Ice Age that humans began to domesticate animals. Archeological evidence in the Near East suggests that wolves were the first animals to enter human societies. Cattle and pigs entered human societies about 9000 years ago and were followed by horses, camels, donkeys, birds, and others around 3000 years ago [5]. Perhaps not surprisingly, animals first entering human societies were valued for the raw materials or agricultural benefits they provided. The earliest recorded legal codes demonstrate the economic value of animals in early human civilization. For example, the Code of Hammurabi from the Mesopotamian civilization of Babylonia is one of the first known recorded legal codes. Its origin dates to around 1750 B.C.E. [6]. The Code consists of 282 rules punishing various behaviors, a significant portion of which relate to the treatment of animals. Many rules specify the recompense owed to the owner of an animal that suffers harm from someone or someone’s animals [7]. 

Early societies responded to bestiality in various ways. In some, such as ancient Greece, sex between humans and nonhuman animals not only went unpunished, but it was possibly commonplace [8]. Other societies, however, punished bestiality severely. The first known bestiality laws were established by another ancient civilization, the Anatolian Hittites, who inscribed their legal code on clay tablets. The tablets, which are estimated to date from 1650 B.C.E. to 1500 B.C.E., also contain an array of rules related to the treatment of animals [9]. Similar to Hammarubi’s code, many Hittite laws treat animals as property with economic value and provide for recompense if one’s animal is injured or killed. Unlike Hammurabi’s code, the Hittites specifically address bestiality in two of their approximately 200 rules. The first rule states:

*If anyone has sexual relations with a pig or a dog, he shall die. He shall bring him to the palace gate. The king may have them killed or he may spare them, but the human shall not approach the king. If an ox leaps on a man, the ox shall die; the man shall not die. They shall substitute one sheep for the man and put it to death. If a pig leaps on a man, it is not an offense* [6] (p. 237).

The second rule states:

*If anyone has sexual relations with either a horse or a mule, it is not an offense, but he shall not approach the king, nor shall he become a priest. If anyone sleeps with an arnuwala-woman, and also sleeps with her mother, it is not an offense* [6] (p. 237).

The Hittite laws regarding bestiality offer a glimpse into the legislative motivation of the Hittite rulers and their society’s views regarding animals in general. It is noteworthy that the laws regarding bestiality do not consider the animal’s economic value. If they did, they would not proscribe death for an ox that “leaps on a man” in sexual excitement. On the contrary, there is some other value system at play defining the animals for which bestiality is punishable. The array of punishments ranges from death for either the human, the animal, or both; to nothing; or to some middling social sanction preventing one from becoming a priest. One legal scholar has suggested that the Hittite society developed rules related to sexual activity with animals based on the concept of cleanliness [1]. Sexual acts thought to put the community at risk of disease were defined as *hurkel* and therefore forbidden.

The texts of modern-day religions provide a window into how other early societies viewed bestiality. The Torah, known as the Pentateuch in the Christian tradition, composes the first five books of the Hebrew and Christian Bibles. The Torah is thought to have been completed by 400 B.C.E. and to express the beliefs and traditions of the Israelites, who lived in the Near East around 600 B.C.E. [10]. One of the books of the Torah, Leviticus, expressly forbids acts of bestiality:

*Do not have sexual relations with an animal and defile yourself with it. A woman must not present herself to an animal to have sexual relations with it; that is a perversion.* (Leviticus 18:23) [11].

Leviticus 20:15 states:

*If a man has sexual relations with an animal, he shall be put to death; and you shall kill the animal* [11].

According to the Israelites, bestiality represented *arayot*, a forbidden relationship. One was meant to give up his or her life rather than engage in such activity. The concern regarding defilement or, in other translations, “becoming unclean” [12], suggests that the Israelites had concerns like the Hittites’ regarding cleanliness or the risk posed to their society by sexual acts involving animals.

Bestiality’s condemnation in the Bible likely affected societal responses to sexual acts involving animals for centuries. As scripture became a source of moral and ethical guidance for followers of Judeo-Christian religions, violations of rules in the Bible could be used as grounds to morally condemn and prosecute such behavior. From the fifteenth to seventeenth centuries in England and mainland Europe, the punishment for “buggery” was death for both human and nonhuman parties [13]. One historian notes that “[t]his disgusting crime appears to have been very common” [14] (p. 148). Such harsh, moralistic punishment was imported to the American colonies. The execution of Thomas Grainger in Plymouth Colony in 1642 serves as a clear example of moral punishment as a motivation for legal action. At the age of 16 or 17, Grainger was sentenced to death for “buggery” involving “a mare, two goats, five sheep, two calves, and a turkey” [15]. The animals were reportedly killed in front of Grainger, then thrown into a pit. Grainger was then hanged. Clearly, whatever moral harms resulted from Grainger’s acts of bestiality were thought to override any potential economic value of the animals with which he had sex.

The religious or moral condemnation of bestiality remains evident as a motivation for legislation in the names of statutes in some jurisdictions of the United States. For example, Maryland’s law is entitled “unnatural or perverted sexual practice” [16]. North Carolina’s law is named “crime against nature” [17] and simply states, “If any person commits a crime against nature, with mankind or beast, he shall be punished as a Class I felon.” The law seemingly criminalizes anal intercourse, including consensual anal intercourse between two adult humans, while failing to address the host of other sexual acts that humans may commit with animals.

### 2.2. The Animal Rights Movement and New Motivations for Animal Maltreatment Legislation

It was not until the 1800′s in Victorian England that the concept that animals have rights separable from those of humans initially developed. At that time, governmental and religious institutions became concerned about the use of animals in activities such as experimentation, transportation, and entertainment such as cockfighting and foxhunting [18]. In 1822, a bill banning cruelty to cattle in England was passed. In 1824, a member of parliament named Thomas Buxton founded the Society for the Prevention of Cruelty to Animals (SPCA), later renamed the Royal Society of Prevention of Cruelty to Animals (RSPCA), which was the first animal welfare charity in the world [19]. The RSPCA was initially focused on the welfare of working animals, including equines that worked in coal mines, and assisted in monitoring animal abuse in public markets, prosecuting cruelty offenders, and promoting the improved treatment of animals to the public at large [20]. Henry Bergh, an American diplomat, witnessed animal abuse in England, as well as the RSPCA’s efforts to address it. After returning to the United States, he incorporated and became the first president of the American Society for the Prevention of Cruelty to Animals (ASPCA) [21]. Bergh was also responsible for the revision of laws regulating the treatment of animals, introducing the term “cruelty” into New York state’s statutory language in the 1860′s and expanding the range of punishable offenses [22]. In response to his activities and successes in New York, other states established their own SPCAs and advocated modifying their own anticruelty statutes [23]. The animal rights approach and Bergh’s efforts are reflected in various laws in the United States that continue to define bestiality under statutes entitled “cruelty to animals,” as in Alaska [24], Colorado [25], and Maine [26].

Most recently, the relationship between animal maltreatment and interpersonal violence, colloquially known as the Link, has served as a motivation for the establishment and expansion of legislation criminalizing bestiality. Though the research base for the Link is limited, particularly when it comes to bestiality, it has had an impact on animal maltreatment legislation. Arkow [4] describes the Link’s role in animal treatment as follows:

*Rather than viewing animal’s well-being as either a matter of moral concern, a result of their alleged inalienable rights, or an incidental activity of municipal code enforcement, what has come to be called “the Link” sees acts of animal abuse as “red flag” potential precursors or predictors of incipient or concurrent antisocial behaviors and interpersonal violence, offering opportunities for earlier intervention* [4] (p. 10).

As evidence of the Link’s impact on animal maltreatment efforts, Arkow cites the incorporation of four different crimes against animals (including bestiality) into the Federal Bureau of Investigation’s National Incident-Based Reporting System (NIBRS), the establishment of laws defining acts of intimidating animal abuse in a relationship context as domestic violence, and the growth of legislation across the country prohibiting bestiality.

Beginning in the 1990′s, legal and ethical scholars began to advance legislative changes consistent with the Link. In 1997, for example, Beirne suggested that bestiality should be reconceptualized as “interspecies sexual assault” because it is coercive, harmful, and often painful and because animals cannot provide consent [27]. In 2005, Otto [28] suggested that the statutes criminalizing acts against animals should mirror the statutes that address similar behavior perpetrated against humans. To that end, he promoted the establishment of “sexual assault of an animal” laws that describe the specific sexual acts, and the relevant organs involved that would constitute an offense. He proposed that an individual would commit a crime if he:


*a. Touches or contacts, or causes an object or another person to touch or contact, the mouth, anus or sex organs of an animal for the purpose of arousing or gratifying the sexual desire of a person; or*
*b. Causes an animal to touch or contact, the mouth, anus or sex organs of an animal for the purpose of arousing or gratifying the sexual desire of a person* [28] (p. 149).

Some states that have recently adopted statutes banning bestiality, such as Hawaii [29] and Oregon [30], utilize Otto’s language of “sexual assault of animal”. In fact, Oregon’s law uses Otto’s recommended text nearly verbatim with the addition of “or animal carcass” to both prongs.

## 3. The Current State of Bestiality Law in the United States

### 3.1. State Statutes

In 2014, I conducted a review of state statutes prohibiting bestiality [3]. At the time, only 31 states had laws criminalizing the behavior, of which 16 imposed a felony and 15 imposed a misdemeanor. Since that time, the legal landscape has changed dramatically. Now all states except New Mexico and West Virginia have statutes that impose sanctions for sexual acts with animals [31]. Twenty-three states impose a misdemeanor, 25 impose a felony, and many states now have felony enhancements for specific sexual acts, for example, coercing a minor to engage in bestiality or for those with prior convictions. Figure 1 is a map of the United States with the jurisdictions defined based on the severity of a conviction for bestiality. Some new laws also require that the individual who is convicted undergo a psychological assessment or face other sanctions, for example, forfeiture of pets or other animals. 

As alluded to above, the titles of state statutes criminalizing bestiality differ significantly, which suggests both the time in which the law was established and the motivation for prohibiting the behavior at that time. Rhode Island’s “abominable and detestable crime against nature” [32] was, predictably, established in 1896. The language of the statute suggests that considerations regarding morality and decency likely motivated the passage of the law. Similarly, Mississippi’s law of “unnatural intercourse” [33] falls under Chapter 29, “Crimes Against Public Morals and Decency,” and, like the previously described North Carolina law, appears to criminalize consensual anal intercourse between adult humans. This law can be traced back to 1930 or earlier. Alternatively, laws categorizing bestiality as animal cruelty or sexual assault tend to be more recent. California amended its law to “sexual contact with animals” in 2019 [34], whereas Hawaii passed its first bestiality law, “sexual assault of an animal” [29], in 2021.

The laws in different jurisdictions across the United States vary in important ways, not just in the title of the statute. Newer laws that utilize modern language tend to provide more granularity regarding the specific acts that are illegal. New Jersey, for example, amended its animal cruelty statute in 2015 to clearly define which sexual behaviors are illegal. The statute states: 

*It shall be unlawful to … use, or cause or procure the use of, an animal or creature in any kind of sexual manner or initiate any kind of sexual contact with the animal or creature, including, but not limited to, sodomizing the animal or creature. As used in this paragraph, “sexual contact” means any contact between a person and an animal by penetration of the penis or a foreign object into the vagina or anus, contact between the mouth and genitalia, or by contact between the genitalia of one and the genitalia or anus of the other. This term does not include any medical procedure performed by a licensed veterinarian practicing veterinary medicine or an accepted animal husbandry practice* [35].

New Jersey’s law is arguably better than those that fail to define unsanctioned behaviors for prosecutorial reasons. First, it clarifies what actual behaviors can result in a charge or conviction. Laws criminalizing the “crime against nature,” on the other hand, contain outdated language that lends itself to various interpretations. Many of these laws conflate bestiality with consensual anal intercourse, clearly targeting male homosexual behavior. Such a connection warrants restructuring of legislation to more specifically focus on acts of bestiality. Second, clear statutory language may assist veterinarians, police, and prosecutors involved in the investigation of bestiality claims. Identification of the relevant organs can aid veterinarians who may have to collect physical evidence in such cases [36]. As Stern and Smith-Blackmore note, “Data obtained from the forensic necropsy should be documented so that it can be viewed or re-created by others and meet the standards required for legal proceedings” [37] (p. 1059), highlighting the potential benefit of legislative specificity for those involved in bestiality investigations.

In addition to criminalizing sexual acts with animals, some state statutes criminalize additional behaviors related to bestiality. An example of this type of legislation is Wisconsin’s statute, amended in 2019. The law defines a variety of behaviors that may be grounds for prosecution, including those that do not involve sexual contact. The law reads:


*No person may knowingly do any of the following:*

*Engage in sexual contact with an animal.*

*Advertise, offer, accept an offer, sell, transfer, purchase, or otherwise obtain an animal with the intent that it be used for sexual contact in this state.*

*Organize, promote, conduct, or participate as an observer of an act involving sexual contact with an animal.*

*Permit sexual contact with an animal to be conducted on any premises under his or her ownership or control.*

*Photograph or film obscene material depicting a person engaged in sexual contact with an animal.*

*Distribute, sell, publish, or transmit obscene material depicting a person engaged in sexual contact with an animal.*

*Possess with the intent to distribute, sell, publish, or transmit obscene material depicting a person engaged in sexual contact with an animal.*

*Force, coerce, entice, or encourage a child who has not attainted the age of 13 years to engage in sexual contact with an animal.*

*Engage in sexual contact with an animal in the presence of a child who has not attained the age of 13 years.*

*Force, coerce, entice, or encourage a child who has attained the age of 13 years but who has not attained the age of 18 to engage in sexual contact with an animal.*
*Engage in sexual contact with an animal in the presence of a child who has attained the age of 13 years but who has not attained the age of 18 years* [38].

The behaviors rendered illegal by Wisconsin’s law are broad. Like child sexual exploitation material (CSEM) laws, the statute criminalizes the dissemination of pornographic material featuring bestiality. In addition, individuals purchasing or selling animals involved in bestiality, organizing or observing bestiality, or permitting bestiality on their premises could all be prosecuted under the law.

Lastly, some state statutes describe additional punishments beyond jail or prison time. Alaskan law, for example, can require those convicted of animal cruelty, including bestiality, to forfeit any affected animals to the state or a custodian, to reimburse the state or custodian for costs related to caring for the animal, and to have their possession of animals prohibited or limited up to ten years [24]. Some states, such as Arizona, have enacted legislation to require individuals convicted of bestiality to undergo psychological assessment and counseling at the convicted person’s expense [39]. This requirement is consistent with the goals of advocates of Link-based law, who contend that animal cruelty is a marker for other forms of interpersonal violence or other antisocial behaviors that can be identified and potentially mitigated through mental health interventions [4]. Finally, some states require offenders to pay a fine, including Louisiana [40], where repeat offenders pay a fine between USD 5000 and USD 25,000.

### 3.2. Federal Legislation

Bestiality has been a crime at the federal level since the 1950′s under the United States Armed Forces Code. The Code states that individuals who engage in “unnatural carnal copulation with an animal” are guilty of bestiality and will be punished through court-martial [41]. In 2019, the U.S. Congress passed the Preventing Animal Cruelty and Torture (PACT) Act. The law criminalizes the creation, sale, and distribution of “crush” videos, or videos depicting cruelty with animals being “crushed, burned, drowned, suffocated, impaled, or otherwise subjected to serious bodily injury” [42] (p. 2) in interstate or foreign commerce. Though some have identified the law as a federal solution that closes a loophole in animal maltreatment legislation [43], others have criticized it as the federalization of what should be state criminal law [44]. 

### 3.3. Law Outside of the United States

The legal status of bestiality around the world is complex. Most nations in the Western world have statutes criminalizing bestiality. Canada modified its statutory definition of bestiality following the high-profile case of R. v. D.L.W. in 2016. D.L.W. was convicted at trial for bestiality for smearing peanut butter on his stepdaughter’s vagina and enticing the family dog to lick it [45]. The Supreme Court overturned the bestiality conviction, however, since his act did not involve actual genital penetration. In response, Senate Bill C-84 was proposed and subsequently received royal assent on 21 June 2019. The law expanded the definition of bestiality to include any contact with an animal for a sexual purpose and expanded punishment to those who commit or compel another to commit bestiality and those who commit bestiality in the presence of a child [46].

In Europe, most nations have laws criminalizing bestiality. Germany, Hungary, Italy, Slovakia, and Slovenia are some exceptions that do not prohibit sexual acts with animals [47]. Among those with bestiality laws, there is wide variability in terms of acts that are punishable and whether other behaviors, such as the distribution or possession of bestiality material, are illegal. The majority Muslim nations have severe penalties for bestiality, including death [48]. Lastly, the status of bestiality in many parts of the world, such as Africa and some nations in Asia, is unclear.

## 4. The Link and the Law

The adoption of the Link as a major motivation for new and expanded bestiality legislation raises some concerns. The research behind the Link, particularly regarding the relationship between bestiality and interpersonal violence and other problematic sexual behavior, is limited. By appraising the quality of the currently available research, one can identify the limitations in relying on such a body of literature to promote legislative reform.

### 4.1. The Link and Bestiality

Considering that humanity has been aware of and legislated bestiality for over 3000 years, it is remarkable how little scientific knowledge exists on the topic. Even general questions regarding the prevalence of the behavior, who engages in it, with what animals, and why remain poorly understood. Though a review of the entire body of research on bestiality is beyond the scope of this article, it is worth mentioning one of the earliest and best studies of bestiality in the general population, which came from the work of sexologist Alfred Kinsey in the 1940′s and 1950′s. At that time, Kinsey reported a relatively high lifetime prevalence of bestiality, particularly in his male subjects [49]. He noted that 8% of men had some form of sexual contact with an animal across their lifespan. That percentage grew to an estimated 40–50% when considering males who were raised on farms. He identified some men who had sex with animals multiple times a week over many years. When assessing women, Kinsey and colleagues found that 1.5% reported a history of bestiality in preadolescence and 3.6% after adolescence [50]. One study of sexual behavior in the general population in the 1970′s identified similar rates of bestiality [51]; however, there have not been recent follow-up studies to assess the current epidemiology of bestiality.

The Link posits that various forms of animal cruelty, including sexual acts with animals, are related to interpersonal violence. In the case of bestiality, then, it is important to understand the risk for violence and other problematic sexual behaviors that is posed by individuals who engage in sex with animals. One of the earliest studies of bestiality and its relationship to other problematic sexual behaviors examined over 500 men who were seeking voluntary evaluation and treatment for paraphilias–disorders of atypical sexual interest–and related behaviors [52]. The authors found that individuals who reported a history of bestiality had, on average, 4.8 comorbid paraphilias, such as pedophilia. This was the third-greatest number of comorbid paraphilias in the sample after individuals who reported a history of obscene phone calls and public masturbation. This finding lends some support to the notion that individuals who engage in bestiality may have a high degree of paraphilic crossover into other problematic sexual behaviors. In another sample of over 44 thousand men and boys undergoing evaluation for sexual misconduct, Abel [53] found that a history of bestiality was the single greatest predictor for the commission of child sexual abuse. Similarly, I assessed all individuals committed as sexually violent predators in the state of Virginia [54] and compared those with a history of bestiality to those without. Those sexually violent predators with a history of bestiality were more likely to report a history of sexually abusing a child and engaging in necrophilic acts, which also lends support to the concept of greater paraphilic crossover in individuals with a history of bestiality.

Some studies have examined the violence histories of prisoners who reported prior acts of bestiality. In one study of 261 prisoners in prisons in the southern United States, researchers found that 6.1% had engaged in bestiality during childhood or adolescence [55]. These individuals were more likely than their peers to have committed an interpersonal crime. They were also more likely to have committed a greater number of interpersonal crimes. In a follow-up survey of 180 prisoners, 12.8% reported a childhood history of bestiality. These offenders were also more likely to have committed interpersonal crimes and a greater number of them [56]. Using the same dataset, Henderson and colleagues [57] compared the history of bestiality to other forms of animal cruelty, such as drowning, hitting, kicking, shooting, or burning animals, and found that the only predictors of recurrent interpersonal violence in adulthood were bestiality and the age at which an individual started abusing animals. They suggested that a childhood history of bestiality may be a precursor for adult interpersonal violence. 

Similarly, some small studies have examined juvenile offenders and their histories of bestiality. Duffield and colleagues [58] evaluated seven boys with a history of bestiality who were referred to a tertiary child and adolescent psychiatry service. They found that four boys had severe conduct disorder, a childhood precursor to antisocial personality disorder. In a survey of 361 adjudicated juvenile offenders, Fleming and colleagues [59] found a prevalence rate of 6% (*n* = 24) of a history of bestiality. Of those, 96% (*n* = 23) admitted to having sexually offended against a human, compared to 52% of the entire sample. Finally, in a study of 32 juvenile sex offenders, Schenk and colleagues [60] found that subjects significantly underreported a history of bestiality when relying on a self-report measure versus a polygraph (37.5% vs. 81.3%). Taken together, these studies suggest that juveniles with a history of sexual offending may be at high risk of engaging in bestiality and vice versa.

Recently, two studies reviewed the criminal histories of adult bestiality offenders. Levitt and colleagues studied 150 adult animal cruelty offenders, of whom 35 were arrested for bestiality [61]. Over one-third of these individuals were also arrested for sexually assaulting a person, which was a significantly greater proportion than those arrested for “active” animal cruelty and animal neglect. More than half of known victims were under the age of 18. In 2019, Edwards published the largest study of bestiality-related arrests [62]. She obtained information on 472 bestiality-related arrests in the United States involving 456 adult offenders between 1975 to 2015. Just over half (52.9%) of subjects had a prior criminal history. Of those, 33.2% had committed a sex offense, 25.7% had committed animal cruelty or bestiality, 15.8% had been convicted of domestic violence, and 10.8% had prior convictions related to CSEM. Fifty arrests involved sex with an animal in addition to the sexual assault of a child or adult. Thirty arrests involved coercion of a child or adult to engage in a sex act with an animal. Levitt and colleagues’ and Edwards’ studies indicate that a potentially large percentage of bestiality offenders—one-third to one-half—have a criminal history, many of whom have a history of sexually offending against humans.

One final area of research relevant to the Link is the relationship between possession of CSEM and interest in bestiality. One study of adult male CSEM offenders found that 15% had collected bestiality material within five years of release from a custodial setting [63]. A recent study assessed the pornography viewing habits of over 250 members of the public and 78 adults previously convicted of CSEM offenses [64]. The study found that 44% of offenders reported viewing adult pornography featuring bestiality, and 18% reported viewing CSEM featuring bestiality, compared to 3% of the public who reported viewing any bestiality material at all. The ratio of pornography viewing between offenders and the public was the greatest for bestiality compared to all other types of pornography, including hentai, teen, rape, and others. This incipient work also supports the concept of paraphilic crossover in individuals interested in bestiality and suggests that CSEM offenders may be more interested in bestiality than members of the general population.

### 4.2. Assessing the Literature

Though compelling, the body of literature summarized above is limited. Perhaps not surprisingly, nearly all studies on the relationship between bestiality and violence or other problematic sexual behaviors involve individuals who have been convicted of crimes or referred for the evaluation of other problematic sexual behaviors. These biased samples make it impossible to understand the risk posed by individuals who engage in sex with animals or have a sexual interest in animals but go undetected. Without an updated epidemiological survey of bestiality in the United States, it is impossible to know the base rate of the behavior and the spectrum of practices involved. It may be that some individuals are exclusive zoophiles whose atypical sexual interests do not extend beyond animals. Abel’s work [52,53] suggests against this hypothesis; however, the individuals in his studies were self-referred or referred by others for a spectrum of problematic sexual behaviors, which would likely fail to sample the hypothesized exclusive zoophile. Because the samples involved in these studies are not representative of the general population—and likely not representative of all individuals who engage in bestiality—their generalizability and use in crafting legislation are significantly curtailed.

An additional problem with studying bestiality is the host of ethical challenges related to such research. In addition to having concerns regarding stigma, individuals with a history of bestiality may be hesitant to discuss acts of bestiality and other problematic sexual behaviors because they are illegal. Participants may fear that their sexual behavior could be reported to legal authorities and that they could be prosecuted. This is likely one reason why much of the latest research on the topic comes from studies in which bestiality was a secondary or incidental finding.

The ideal study to assess the relationship between bestiality and interpersonal violence, and other problematic sexual behaviors would evaluate a cohort of children, adolescents, and adults who have had sex with animals and a control cohort that has not. At various timepoints in these individuals’ lives, researchers could assess individuals’ history of sexual behavior, violent behavior, and legal involvement [65]. Such a study could shed light on the differences in rates of interpersonal violence, sexual offending, and relevant mental health outcomes between individuals with a history of bestiality and those without. The expense, time involved, large study sample needed, and difficulty identifying individuals willing to disclose a history of bestiality render it unlikely that such a study will be conducted.

### 4.3. Research and the Law

Given the limitations of research on the Link between bestiality and interpersonal violence, it is unclear that such research should motivate new laws and the remodeling of old laws. In fact, it is questionable whether there should even be a need for Link-related studies to support such legislative action. By arguing that animal cruelty matters and deserves attention because of the risk posed to humans by offenders, one may, in effect, be arguing that violence against animals is not worthy of legislative action by itself. Furthermore, the research basis for arguing that bestiality’s link to interpersonal offending is so strong that it warrants legal intervention is simply not present. At best, the research suggests that individuals with interest in bestiality likely have a host of atypical sexual interests, some of which may lead to interpersonal offending or the use of problematic pornography. A failure to recognize and acknowledge the limitations of the research undergirding the Link poses a risk of overstating research results that could lead to legislative decisions.

Despite these problems, encouraging the development and expansion of bestiality legislation by promising potential benefits to humans may be the most convincing and expedient way to achieve such ends. At the same time, it would be beneficial to shape legislation that assists in improving our understanding of bestiality. To that end, the mandated psychological assessments of bestiality offenders introduced in some states may prove useful. Such evaluations could add to the knowledge base by clarifying the problematic sexual behaviors most associated with bestiality and the risk of interpersonal violence posed by offenders. The information derived from such assessments could also assist in identifying the rate of genuine sexual interest in animals, or zoophilia, that exists in individuals who engage in and are arrested for bestiality. More generally, mandated evaluations could further define the various motivations for which individuals have sex with animals.

## 5. Conclusions

Despite societies’ efforts to address bestiality through legal mechanisms since the time of the Hittites, legislative efforts have not been guided by a scientific understanding of the behavior and the risk posed by individuals who have sex with animals. Statutes prohibiting bestiality have changed in response to evolving social and moral considerations over the centuries, but research efforts have lagged far behind. The concept of the Link between animal cruelty and interpersonal violence has recently served as a major motivation for establishing and updating anticruelty statutes, including those that criminalize bestiality. The research basis for this aspect of the Link is tenuous, however. At best, the literature suggests that individuals who engage in bestiality are likely to have other atypical sexual interests that can place humans, including children, at risk. It may be useful to consider ways in which laws can help expand our knowledge about bestiality, for example, through mandated psychological assessments of offenders. Data derived from such assessments could clarify the behaviors and motivations of individuals who engage in bestiality, as well as their history of violence and other problematic sexual behaviors or violence. Such information could serve to bolster a broader understanding of bestiality and elucidate the appropriate mechanisms by which to manage offenders.

## Figures and Tables

**Figure 1 animals-12-01525-f001:**
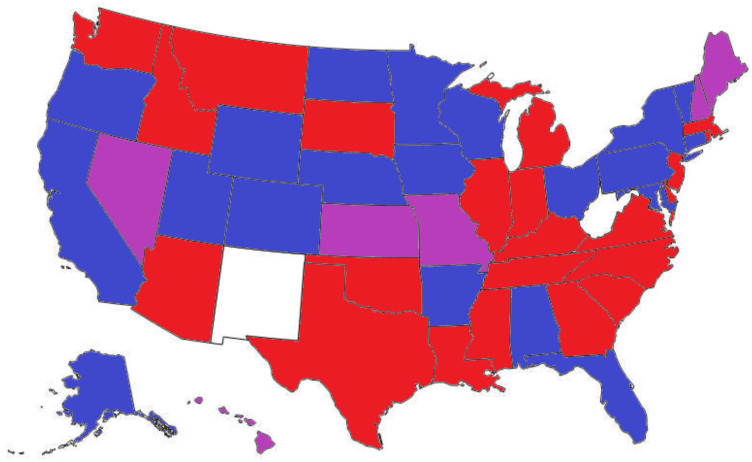
Jurisdictions that prohibit bestiality in the United States. States imposing a misdemeanor in blue, a misdemeanor with felony enhancements in purple, and a felony in red.

## Data Availability

All data discussed in this article are publicly available from other sources.

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
