# Peer review of "Bestiality Law in the United States: Evolving Legislation with Scientific Limitations"

_animals, 2022, doi:10.3390/ani12121525_

Round 1

Reviewer 1 Report

This is an interesting paper that covers a rarely discussed subject for a mainstream animal research publication. The observations on the emphasis placed on ‘The Link’ as a catalyst for legislative reform were eye-opening, and the realisation that what the Link may do is undermine the drive for animal rights and for legislation that values animals as more than mere property, or indeed ‘means to an end,’ is fascinating, especially as legislation is moving towards recognition of animal sentience.

I would have liked to see more discussion on bestiality as an act of violence against animals, rather than as a sexual act - I am not sure that this was covered sufficiently. This could then lead in to the Link discussion, with perhaps two distinct strands – the link between bestiality and paraphilia, and that between bestiality and other violent acts. I was going to mention the topic of rape and whether it is a sexual act or an act of violence, but I am aware that this is a huge subject, and perhaps would derail the focus of the paper.

Some very minor points below:

P4, line 180 – I wonder if the phrase ‘statutes that sanction’ could be replaced with ‘statutes that impose sanctions’?  I find the use of the verb ‘sanction’ very confusing as it can have two opposite meanings, whereas ‘imposing sanctions’ is clearly referring to a criminal act.

P5, line 205 – ‘specificity regarding the specific acts’ might be better phrased as ‘clarity (or granularity) regarding the specific acts’?

P7, lines 307-309. The sentence beginning ‘In the case of sexual acts with animals…’ does not make sense. I think there needs to be a ‘that’ between ‘behaviors’ and ‘is posed’.

P9, line 389 – when listing the potential problems with the proposed research, I feel that research ethics should be included?

Author Response

Reviewer 1:

This is an interesting paper that covers a rarely discussed subject for a mainstream animal research publication. The observations on the emphasis placed on ‘The Link’ as a catalyst for legislative reform were eye-opening, and the realisation that what the Link may do is undermine the drive for animal rights and for legislation that values animals as more than mere property, or indeed ‘means to an end,’ is fascinating, especially as legislation is moving towards recognition of animal sentience.

Thank you for your positive feedback.

I would have liked to see more discussion on bestiality as an act of violence against animals, rather than as a sexual act - I am not sure that this was covered sufficiently. This could then lead in to the Link discussion, with perhaps two distinct strands – the link between bestiality and paraphilia, and that between bestiality and other violent acts. I was going to mention the topic of rape and whether it is a sexual act or an act of violence, but I am aware that this is a huge subject, and perhaps would derail the focus of the paper.

Thank you for your comment. I agree that this is an important distinction and attempted to separate out these issues in different paragraphs in the section summarizing the literature. Unfortunately, there is limited data on the relationship between bestiality and general violence risk, so there is little utility in separating out the two topics.

Some very minor points below:

P4, line 180 – I wonder if the phrase ‘statutes that sanction’ could be replaced with ‘statutes that impose sanctions’?  I find the use of the verb ‘sanction’ very confusing as it can have two opposite meanings, whereas ‘imposing sanctions’ is clearly referring to a criminal act.

I made this change.

P5, line 205 – ‘specificity regarding the specific acts’ might be better phrased as ‘clarity (or granularity) regarding the specific acts’?

I made this change.

P7, lines 307-309. The sentence beginning ‘In the case of sexual acts with animals…’ does not make sense. I think there needs to be a ‘that’ between ‘behaviors’ and ‘is posed’.

You are correct. I added “that” to this sentence.

P9, line 389 – when listing the potential problems with the proposed research, I feel that research ethics should be included?

I have added a discussion of ethical considerations in research to this section.

Reviewer 2 Report

Although this is a potentially useful overview of legal response to bestiality, it contains several errors and overlooks a considerable amount of previously published material key to understanding this topic.

1. The section on the history of bestiality and bestiality law ignores several key references on the subject:

Evans, E.P. 1906 (reprinted 1987). The Criminal Prosecution and Capital Punishment of Animals. London: Faber and Faber.

Fudge, E.(2000). Monstrous acts: Bestiality in early modern England. History Today, 50, 20-25.

Garrard, G. (2017). Bestial humans and sexual animals: Zoophilia in law and literature. In M. Lundblad (Ed.). Animalities: Literary and cultural studies beyond the human (pp. 211-235). Edinburgh University Press.

Miletski, H. (2005). A history of bestiality. In A.M. Beetz. & A.L. Podberscek (Eds.). Bestiality and zoophilia: Sexual relations with animals (pp. 1-22). Purdue University Press.

2. The paragraph describing the origins of the ASPCA incorrectly says that Bergh was “an American diplomat based in England” (line 133). He was based in Russia but met with RSPCA founders on his travels back to America. The cited Furstinger pop biography is unreliable. More recent and accurate coverage of ASPCA history is provided in:

Lane, M.S. and Zawistowski, S.L. (2008). Heritage of care: The American Society for the Prevention of Cruelty to Animals. Westport, CT: Praeger.

Freeberg, E. (2020). A Traitor to His Species: Henry Bergh and the Birth of the Animal Rights Movement. By Ernest Freeberg.  Basic Books

3. The assertion (line 148) “The research base for the Link is weak..” is argumentative, controversial and highly questionable. The most recent Link bibliography (www.animaltherapy.net) contains over 1,800 citations supportive of the concept, with the majority being from refereed works in a wide variety of fields. The assertion that the link is weak “particularly when it comes to bestiality”  is at odds with the author’s later statement  (line 352) that a potentially large percentage of bestiality offenders- one third to one-half- have a criminal history. The assertion that “most” do not have a history of sex offenses against humans is technically true since the % in Edwards was 33.2%, but that is hardly insubstantial.

4. The discussion of the use of the term “sexual assault of an animal” (line 172) implies the origin of the term is from Otto (2005). Credit for introducing this concept in animal law should go to the detailed discussion of changing the concept of bestiality to one of animal sexual assault by the significant work of Beirne:

Beirne, P. (1997). Rethinking Bestiality: Towards a concept of interspecies sexual assault. Theoretical Criminology, 1(3), 317-340.

Beirne, P., Maher, J., & Pierpoint, H. (2017). Animal sexual assault. In J. Maher, H. Pierpoint, & P. Beirne (eds.). The Palgrave International Handbook of Animal Abuse Studies. London: Palgrave Macmillan, pp 59-85.

5. The section describing the antiquarian language inherent in many state laws on bestiality, starting at line 192, would benefit from mentioning that several states link sex with animals with homosexuality: Idaho §18-6605 prohibits “The infamous crime against nature, committed with mankind or any animal”; Massachusetts §34-272 and Oklahoma §21.886 prohibit “the abominable and detestable crime against nature, either with mankind or with a beast”; Michigan’s Penal Code §750.158 prohibits “the abominable and detestable crime against nature either with mankind or with any animal”; South Carolina is even more archaic with §16-15-120 prohibiting “the abominable crime of buggery, whether with mankind or with beast.” It is such connections that has motivated much of the restructuring of bestiality laws to more specifically focus on animal sexual assault.

6. The suggestion (line 221) that “Identification of the relevant organs can aid veterinarians who may have to collect physical evidence in such cases”  should further reference some of the substantial existing forensic veterinary guidelines on investigating and documenting animal sexual assault:

Bradley, N., & Rasile, K. K. (2014, April). Recognition and management of animal sexual abuse. Clinician’s Brief, 73-75.

Merck, M. D., & Miller, D.M. (2013). Sexual abuse. In M. Merck (Ed.). Veterinary forensics: Animal cruelty investigations, 2nd ed. (pp. 233-241). New York: Wiley.

Munro, H.M.C. and M.V. Thrusfield. (2001). ‘Battered pets’: sexual abuse. Journal of Small Animal Practice, 42:333-337.

Munro, H.M.C. (2006). Animal sexual abuse: A veterinary taboo? The Veterinary Journal 172(2), 195-197.

Stern, A.W., & Smith-Blackmore, M. (2016). Veterinary forensic pathology of animal sexual abuse. Veterinary Pathology, 53(5), 1057-1066.

7. The section on Law outside of the United States (starting at line 277) overlooks several key references:

Vetter, S., Boros, A., & Ózsvári, L. (2020). Penal sanctioning of zoophilia in light of the legal status of animals: A comparative analysis of fifteen European countries. Animals, 10, 1024; doi:10.3390/ani10061024

Canadian Centre for Child Protection Inc., "Bestiality" as reflected in Canadian case law, CanLII Authors Program, 2018 CanLIIDocs 266, <https://canlii.ca/t/t0dx>

8. The literature review on the Link and bestiality, while including some of the more significant work (eg.  Levitt et al. and Edwards, overlooks many other works examining this connection, among them:

Ascione, F.R. (2005). Bestiality: Petting, ‘humane rape,’ sexual assault, and the enigma of sexual interactions between humans and non-human animals.” In A.M. Beetz. & A.L. Podberscek (Eds.). Bestiality and zoophilia: Sexual relations with animals (pp. 120-129). West Lafayette, IN: Purdue University Press.

Beetz, A. M. & Podberscek, A. L. (eds.) (2005). Bestiality and Zoophilia: Sexual Relations with Animals. West Lafayette, IN: Purdue University Press

Beetz, A.M. (2008). Bestiality and zoophilia: A discussion of sexual contact with animals. In F.R. Ascione (Ed.). International handbook of animal abuse and cruelty: Theory, research, and application (pp. 201-220). West Lafayette, IN: Purdue University Press.

Beetz, A. M. (2005). Bestiality and zoophilia: Associations with violence and sex offending. In A.M. Beetz. & A.L. Podberscek (Eds.). Bestiality and zoophilia: Sexual relations with animals (pp. 46-70). West Lafayette, IN: Purdue University Press.

Beetz, A. M. (2005). New insights into bestiality and zoophilia. In A.M. Beetz. & A.L. Podberscek, eds.: Bestiality and zoophilia: Sexual relations with animals (pp. 98-119). West Lafayette, IN: Purdue University Press.

Duffield, G., Hassiotis, A., & Vizard, E. (1998). Zoophilia in young sexual abusers. Journal of Forensic Psychiatry 9, 294-304.

Fleming, W.M., Jory, B. & Burton, D.L. (2002). Characteristics of juvenile offenders admitting to sexual activity with nonhuman animals. Society & Animals,10(1), 31-45.

Sandnabba, N. K., Santtila, P., Beetz, A. M., Nordling, N., & Alison, L. (2002). Characteristics of a sample of sadomasochistically-oriented males with recent experience of sexual contact with animals. Deviant Behavior, 23(6), 511–530.

Schenk, A. M., Cooper-Lehki, C., Keelan, C. M., & Fremouw, W. J. (2014). Underreporting of bestiality among juvenile sex offenders: Polygraph versus selfreport. Journal of Forensic Sciences, 59(2), 540–542

Tallichet, S.E. & Hensley, C. (2013). Animal cruelty and sexual deviance. In M.P. Brewster & C.L. Reyes, eds.: Animal cruelty: A multidisciplinary approach to understanding (pp. 181-195). Durham, NC: Carolina Academic Press.

9. Many animal advocates agree with the position that strengthening and enforcing animal protection laws in the interests of animals themselves as sentient creatures (or at least “sentient property”) does not require dependence on the Link. However, the legislative reality has been that animal protection legislation has been of little interest to most legislatures unless there is a perceived benefit to humans. This is particularly true of legislation related to bestiality. if the author has not attended a hearing where bestiality is raised… he should. The aversion to any discussion of bestiality in a political forum is a significant deterrent to any progress unless there is the perception of some benefit to people in addressing the issue. The idea that there would be any interest in advocating  for legislation that improves our understanding of bestiality is, at best, naïve.

Author Response

Reviewer 2:

Although this is a potentially useful overview of legal response to bestiality, it contains several errors and overlooks a considerable amount of previously published material key to understanding this topic.

  1. The section on the history of bestiality and bestiality law ignores several key references on the subject:

Evans, E.P. 1906 (reprinted 1987). The Criminal Prosecution and Capital Punishment of Animals. London: Faber and Faber.

Fudge, E.(2000). Monstrous acts: Bestiality in early modern England. History Today, 50, 20-25.

Garrard, G. (2017). Bestial humans and sexual animals: Zoophilia in law and literature. In M. Lundblad (Ed.). Animalities: Literary and cultural studies beyond the human (pp. 211-235). Edinburgh University Press.

Miletski, H. (2005). A history of bestiality. In A.M. Beetz. & A.L. Podberscek (Eds.). Bestiality and zoophilia: Sexual relations with animals (pp. 1-22). Purdue University Press.

Thank you for these recommendations. I have incorporated information from them into the manuscript.

  1. The paragraph describing the origins of the ASPCA incorrectly says that Bergh was “an American diplomat based in England” (line 133). He was based in Russia but met with RSPCA founders on his travels back to America. The cited Furstinger pop biography is unreliable. More recent and accurate coverage of ASPCA history is provided in:

Lane, M.S. and Zawistowski, S.L. (2008). Heritage of care: The American Society for the Prevention of Cruelty to Animals. Westport, CT: Praeger.

Freeberg, E. (2020). A Traitor to His Species: Henry Bergh and the Birth of the Animal Rights Movement. By Ernest Freeberg.  Basic Books

Thank you for pointing this out. I have modified the text and the reference accordingly.

  1. The assertion (line 148) “The research base for the Link is weak..” is argumentative, controversial and highly questionable. The most recent Link bibliography (www.animaltherapy.net) contains over 1,800 citations supportive of the concept, with the majority being from refereed works in a wide variety of fields. The assertion that the link is weak “particularly when it comes to bestiality”  is at odds with the author’s later statement  (line 352) that a potentially large percentage of bestiality offenders- one third to one-half- have a criminal history. The assertion that “most” do not have a history of sex offenses against humans is technically true since the % in Edwards was 33.2%, but that is hardly insubstantial.

I have modified the wording of the phrase regarding the research base, as well as that summarizing the Edwards finding.

  1. The discussion of the use of the term “sexual assault of an animal” (line 172) implies the origin of the term is from Otto (2005). Credit for introducing this concept in animal law should go to the detailed discussion of changing the concept of bestiality to one of animal sexual assault by the significant work of Beirne:

Beirne, P. (1997). Rethinking Bestiality: Towards a concept of interspecies sexual assault. Theoretical Criminology, 1(3), 317-340.

Beirne, P., Maher, J., & Pierpoint, H. (2017). Animal sexual assault. In J. Maher, H. Pierpoint, & P. Beirne (eds.). The Palgrave International Handbook of Animal Abuse Studies. London: Palgrave Macmillan, pp 59-85.

This is an excellent point. I have added information to this section and cited Beirne.

  1. The section describing the antiquarian language inherent in many state laws on bestiality, starting at line 192, would benefit from mentioning that several states link sex with animals with homosexuality: Idaho §18-6605 prohibits “The infamous crime against nature, committed with mankind or any animal”; Massachusetts §34-272 and Oklahoma §21.886 prohibit “the abominable and detestable crime against nature, either with mankind or with a beast”; Michigan’s Penal Code §750.158 prohibits “the abominable and detestable crime against nature either with mankind or with any animal”; South Carolina is even more archaic with §16-15-120 prohibiting “the abominable crime of buggery, whether with mankind or with beast.” It is such connections that has motivated much of the restructuring of bestiality laws to more specifically focus on animal sexual assault.

I have added information to emphasize this point in the paragraph discussing problems with archaic legal language.

  1. The suggestion (line 221) that “Identification of the relevant organs can aid veterinarians who may have to collect physical evidence in such cases”  should further reference some of the substantial existing forensic veterinary guidelines on investigating and documenting animal sexual assault:

Bradley, N., & Rasile, K. K. (2014, April). Recognition and management of animal sexual abuse. Clinician’s Brief, 73-75.

Merck, M. D., & Miller, D.M. (2013). Sexual abuse. In M. Merck (Ed.). Veterinary forensics: Animal cruelty investigations, 2nd ed. (pp. 233-241). New York: Wiley.

Munro, H.M.C. and M.V. Thrusfield. (2001). ‘Battered pets’: sexual abuse. Journal of Small Animal Practice, 42:333-337.

Munro, H.M.C. (2006). Animal sexual abuse: A veterinary taboo? The Veterinary Journal 172(2), 195-197.

Stern, A.W., & Smith-Blackmore, M. (2016). Veterinary forensic pathology of animal sexual abuse. Veterinary Pathology, 53(5), 1057-1066.

Thank you for this recommendation. I have added some relevant veterinary references.

  1. The section on Law outside of the United States (starting at line 277) overlooks several key references:

Vetter, S., Boros, A., & Ózsvári, L. (2020). Penal sanctioning of zoophilia in light of the legal status of animals: A comparative analysis of fifteen European countries. Animals, 10, 1024; doi:10.3390/ani10061024

Canadian Centre for Child Protection Inc., "Bestiality" as reflected in Canadian case law, CanLII Authors Program, 2018 CanLIIDocs 266, <https://canlii.ca/t/t0dx>

These are excellent suggestions and I have expanded the section on law outside of the United States to incorporate them.

  1. The literature review on the Link and bestiality, while including some of the more significant work (eg.  Levitt et al. and Edwards, overlooks many other works examining this connection, among them:

Ascione, F.R. (2005). Bestiality: Petting, ‘humane rape,’ sexual assault, and the enigma of sexual interactions between humans and non-human animals.” In A.M. Beetz. & A.L. Podberscek (Eds.). Bestiality and zoophilia: Sexual relations with animals (pp. 120-129). West Lafayette, IN: Purdue University Press.

Beetz, A. M. & Podberscek, A. L. (eds.) (2005). Bestiality and Zoophilia: Sexual Relations with Animals. West Lafayette, IN: Purdue University Press

Beetz, A.M. (2008). Bestiality and zoophilia: A discussion of sexual contact with animals. In F.R. Ascione (Ed.). International handbook of animal abuse and cruelty: Theory, research, and application (pp. 201-220). West Lafayette, IN: Purdue University Press.

Beetz, A. M. (2005). Bestiality and zoophilia: Associations with violence and sex offending. In A.M. Beetz. & A.L. Podberscek (Eds.). Bestiality and zoophilia: Sexual relations with animals (pp. 46-70)West Lafayette, IN: Purdue University Press.

Beetz, A. M. (2005). New insights into bestiality and zoophilia. In A.M. Beetz. & A.L. Podberscek, eds.: Bestiality and zoophilia: Sexual relations with animals (pp. 98-119)West Lafayette, IN: Purdue University Press.

Duffield, G., Hassiotis, A., & Vizard, E. (1998). Zoophilia in young sexual abusers. Journal of Forensic Psychiatry 9, 294-304.

Fleming, W.M., Jory, B. & Burton, D.L. (2002). Characteristics of juvenile offenders admitting to sexual activity with nonhuman animals. Society & Animals,10(1), 31-45.

Sandnabba, N. K., Santtila, P., Beetz, A. M., Nordling, N., & Alison, L. (2002). Characteristics of a sample of sadomasochistically-oriented males with recent experience of sexual contact with animals. Deviant Behavior, 23(6), 511–530.

Schenk, A. M., Cooper-Lehki, C., Keelan, C. M., & Fremouw, W. J. (2014). Underreporting of bestiality among juvenile sex offenders: Polygraph versus selfreport. Journal of Forensic Sciences, 59(2), 540–542

Tallichet, S.E. & Hensley, C. (2013). Animal cruelty and sexual deviance. In M.P. Brewster & C.L. Reyes, eds.: Animal cruelty: A multidisciplinary approach to understanding (pp. 181-195)Durham, NC: Carolina Academic Press.

Thank you for these recommended references. I have incorporated information from many of them into the article. I would note, however, that many of the references above are not empirical studies and/or are summaries of articles on self-identified zoophiles without any discussion of violence or legal troubles. These references are therefore irrelevant to a discussion of Link literature.

  1. Many animal advocates agree with the position that strengthening and enforcing animal protection laws in the interests of animals themselves as sentient creatures (or at least “sentient property”) does not require dependence on the Link. However, the legislative reality has been that animal protection legislation has been of little interest to most legislatures unless there is a perceived benefit to humans. This is particularly true of legislation related to bestiality. if the author has not attended a hearing where bestiality is raised… he should. The aversion to any discussion of bestiality in a political forum is a significant deterrent to any progress unless there is the perception of some benefit to people in addressing the issue. The idea that there would be any interest in advocating for legislation that improves our understanding of bestiality is, at best, naïve.

Thank you. Your point is well-taken. I have modified the beginning of the second-to-last paragraph to specifically address this issue.

Reviewer 3 Report

You have written an interesting paper. I would have liked to see section 3.3 to be expanded to include more information about other countries. Please note spelling of country Hungary.

Author Response

Reviewer 3:

You have written an interesting paper. I would have liked to see section 3.3 to be expanded to include more information about other countries. Please note spelling of country Hungary.

Thank you for your positive feedback. I edited the spelling of “Hungary” and added some additional information about other countries.

Round 2

Reviewer 2 Report

This version is substantially improved. You have paid close attention to the suggestions that were made.

Reviewer 3 Report

Well done, this paper is now much improved.